# Gut Microbiome Changes in Patients with Active Left-Sided Ulcerative Colitis after Fecal Microbiome Transplantation and Topical 5-aminosalicylic Acid Therapy

**DOI:** 10.3390/cells9102283

**Published:** 2020-10-13

**Authors:** Dagmar Schierová, Jan Březina, Jakub Mrázek, Kateřina Olša Fliegerová, Simona Kvasnová, Lukáš Bajer, Pavel Drastich

**Affiliations:** 1Institute of Animal Physiology and Genetics of the Czech Academy of Science, v.v.i., 142 20 Prague, Czech Republic; fliegerova@iapg.cas.cz (K.O.F.); kvasnova@iapg.cas.cz (S.K.); 2Hepatogastroenterology Department, Institute for Clinical and Experimental Medicine, 140 21 Prague, Czech Republic; brej@ikem.cz (J.B.); lukasbajer1@gmail.com (L.B.); padr@ikem.cz (P.D.)

**Keywords:** ulcerative colitis, microbiome, fecal microbiome transplantation, 5-ASA

## Abstract

Ulcerative colitis (UC) is an inflammatory bowel disease, and intestinal bacteria are implicated in the pathogenesis of this disorder. The administration of aminosalicylates (5-ASA) is a conventional treatment that targets the mucosa, while fecal microbial transplantation (FMT) is a novel treatment that directly targets the gut microbiota. The aim of this study was to identify changes in fecal bacterial composition after both types of treatments and evaluate clinical responses. Sixteen patients with active left-sided UC underwent enema treatment using 5-ASA (*n* = 8) or FMT (*n* = 8) with a stool from a single donor. Fecal microbiota were analyzed by 16S rDNA high-throughput sequencing, and clinical indices were used to assess the efficacy of treatments. 5-ASA therapy resulted in clinical remission in 50% (4/8) of patients, but no correlation with changes in fecal bacteria was observed. In FMT, remission was achieved in 37.5% (3/8) of patients and was associated with a significantly increased relative abundance of the families Lachnospiraceae, Ruminococcaceae, and Clostridiaceae of the phylum Firmicutes, and Bifidobacteriaceae and Coriobacteriaceae of the phylum Actinobacteria. At the genus level, *Faecalibacterium*, *Blautia*, *Coriobacteria*, *Collinsela*, *Slackia*, and *Bifidobacterium* were significantly more frequent in patients who reached clinical remission. However, the increased abundance of beneficial taxa was not a sufficient factor to achieve clinical improvement in all UC patients. Nevertheless, our preliminary results indicate that FMT as non-drug-using method is thought to be a promising treatment for UC patients.

## 1. Introduction

Ulcerative colitis (UC) is a type of inflammatory bowel disease (IBD) characterized by chronic inflammation of the large intestine with remitting periods of relative quiescence and periods of mild to severe flares that affect the patient’s quality of life substantially. Diarrhea mixed with blood accompanied with abdominal pain are the main symptoms. A sharply rising prevalence of this disease has been seen in developed countries, but a rapidly increasing incidence in newly industrialized countries is also evident. Its global prevalence is predicted to affect up to 30 million individuals by 2025 [1]. The precise etiology of UC is not clear; however, it is well documented that UC patients suffer from intestinal disturbance, reduced species diversity and richness, increased gut mucosa permeability, and hampered immune response that manifests in an inflammatory milieu of the patients’ colon [2]. Although it is still not known whether the dysbiosis is a cause or consequence of the disease, one possible way to revert this dysbiotic state to healthy homeostasis is a fecal microbiome transplantation (FMT), which is still considered an alternative method of treatment.

The first treatment option for mild or moderate UC that helps to reduce inflammation are aminosalicylates, specifically 5-aminosalicylic acid (5-ASA). This drug can be used as a short-term treatment for flare-ups but can also be taken long-term to maintain remission. Most of the time they are only administered orally, and thus do not reach their full potential in the colon [3]. The effect of 5-ASA on the intestinal microbial community is usually studied in mucosa, which is a target site of action. Xu et al. [4] demonstrated a higher abundance of Firmicutes and lower levels of Proteobacteria in the inflamed mucosa of 5-ASA-treated patients. Olaisen et al. [5] determined that the mucosal 5-ASA concentration was positively associated with mucosal bacterial diversity and bacterial compositions. A high mucosal 5-ASA concentration was also found to be related to the reduced abundance of pathogenic bacteria such as Proteobacteria and increased abundance of several favorable families (Lachnospiraceae and Ruminococcaceae) and genera such as *Faecalibacterium* and *Coprococcus*. 5-ASA may have beneficial effects on the mucosal microbiome, with high concentrations altering dysbiosis in UC. Olaisen et al. [5] highlighted that mucosal 5-ASA concentration is associated with changes in mucosal bacterial composition, however, the fecal microbiota was not changed to the same extent.

As opposed to 5-ASA treatment, FMT is directly focused on changing the composition of the intestinal microbiota. Emerging evidence is proving an important effect of the human gut microbiota on health, as well as the involvement of the intestinal bacteria in several diseases. Patients with UC indeed have a different microbial community in their colon compared to healthy subjects, and specific members of the intestinal microbiota were found to be dramatically affected. Typically, UC is characterized by decreased Firmicutes, in particular, beneficial butyrate-producing bacteria are diminished while members of Bacteroidetes and Proteobacteria are increased, with this being associated with disease relapse [6,7]. The goal of FMT in UC patients is therefore to achieve greater bacterial diversity; to accomplish a higher bacterial similarity of the recipient to the donor; to introduce beneficial taxa; and, most importantly, to establish a new less inflammation-prone community in the recipient colon. The main aim is of course the remission of disease. FMT was shown to be highly effective in the treatment of *Clostridium difficile* infection [8,9], and in UC patients it seems to be a promising new therapeutic option, which, however, to date has not achieved results as good as *C. difficile* treatment [10].

Several studies have already demonstrated a positive influence of FMT and its potential therapeutic value for the treatment of UC. Significantly increased bacterial diversity was described in the majority of studies [11,12,13,14,15], however, it is not the only prerequisite for successful FMT, as Kump et al. [16] and Damman et al. [17] did not find alpha diversity differences but still observed temporary disease remission in some patients. A significantly increased similarity of the patient to the donor was described by several authors [10,15,16,18], indicating a high rate of microbiota transfer and shift of the bacterial community to a new, healthier composition. FMT resulted in increased levels of certain beneficial taxa such as *Faecalibacterium prausnitzii* and other butyrate-producing bacteria belonging mainly to the families Lachnospiraceae and Ruminococcaceae of the phylum Firmicutes [12,16,19,20]. The loss of potentially harmful taxa such as adherent-invasive *Escherichia coli* [21] and decreased abundance of members from the family Enterococcaceae [6,16,20] is another positive impact of FMT. Already, more than 30 years have passed from the first documented FMT in a UC patient [22], yet we still lack details about choosing a suitable donor, the mode of FMT application, and the reasons why some UC patients do not respond to FMT or relapse, even after initial remission induced by FMT [10]. Research in this field is therefore of high importance.

In this study, we aimed to determine the influence of 5-ASA topical treatment and FMT treatment on the fecal bacterial community in patients with left-sided UC and to evaluate the consequent clinical response. Patients with active left-sided UC were chosen due to the favorable inflammation localization, which is a site that can be easily reached by enema. Sixteen UC patients during the therapy provided 60 stool samples, which were analyzed using a high-throughput sequencing (HTS) approach for bacterial diversity at various phylogenetic levels. The preliminary results of this work endeavor to contribute to the scientific debate about the effectiveness of FMT for UC patients and elucidating its non-responsiveness in some subjects.

## 2. Materials and Methods

### 2.1. Patients, Donor, and Study Design

A total of 60 samples were collected from 16 outpatients who met the UC diagnostic criteria on the basis of typical clinical, endoscopic, and histologic findings carried out at the hepatogastroenterology clinic of the Institute for Clinical and Experimental Medicine (Prague, Czech Republic). Patients were randomized into two groups, FMT (*n* = 8) and 5-ASA (*n* = 8), according to the type of treatment. The treatment regimen of the FMT group consisted of an enema prepared from 50 g of donor stool dissolved in 150 mL of saline solution administered 5 times in the 1st week, then once a week until the end of the 6th week. The stool for FMT was prepared from multiple samples originating from 1 donor collected before the start of the study. Fresh stool was weighed, diluted in physiological solution, and homogenized in a household blender. The homogenate was then twice filtered through gauze and mixed well with 17 g of pharmaceutical-grade glycerol. The suspension was aspirated into 200 cc syringes and stored at −80 °C. This donor stool preparation was defrosted for 1 hour at room temperature and completely thawed at 37 °C in water bath prior to application. The treatment regimen of the 5-ASA group consisted of an enema with 4 g of mesalazine (5-ASA) administered daily for 2 weeks, then every 2nd day until the end of the 6th week. Fecal samples were collected before the start of treatment; during the treatment on weeks 2, 4, and 6 (24 h prior to the enema application); and after the treatment on week 12. Sample collection was not fully complete however, each patient provided at least 1 sample before and 1 sample during the treatment (Appendix A). The stool donor was a healthy middle-aged man (32 years old) with a BMI (body mass index) of 23.8 who was not related to any of the patients. The donor’s medical and surgical history was obtained and showed no history of infectious, autoimmune, and gastrointestinal disease; chronic diseases or allergies; drug or chemotherapy use; antibiotic therapy within the past 6 months; or hospitalization in the last 3 months. The donor underwent laboratory evaluation including blood testing (complete blood count), C-reactive protein test, erythrocyte sedimentation rate test, biochemical tests for viral disease, and stool testing for infectious bacteria and parasites. A high-throughput sequencing of 16S rDNA of the donor’s stool sample was performed before FMT treatment to exclude the presence of undesirable/harmful bacteria. The patients and donor were informed about the potential risks and benefits of FMT, and all participants in the experiment gave their written informed consent to the protocol, which was approved by the Ethics Committee of Institute of Clinical and Experimental Medicine and Thomayer Hospital (NCT03104036).

### 2.2. DNA Extraction

Stool samples from patients and the donor were frozen and stored at −80 °C, and subsequently approximately 1 g of each sample was freeze-dried (LYOVAC GT 2, Leybold Heraeus). Genomic DNA was extracted from lyophilized samples using the method of Yu and Morrison [23], combining rapid beating in a FastPrep-24 homogenizer (MP Biomedicals) with purification in QIAamp DNA Stool Mini Kit columns (Qiagen). The concentration and purity of extracted nucleic acids were checked using a NanoDrop 2000c UV–VIS spectrophotometer (Thermo Scientific). DNA extracts were stored at −20 °C until their use.

### 2.3. PCR Amplification and High-throughput Sequencing

The amplification of the bacterial variable V4-V5 region of 16S rRNA was performed according to Fliegerova et al. [24] using EliZyme HS Robust MIX Red (Elisabeth Pharmacon) and 10 μM of each primer (forward: GGATTAGATACCCTGGTAGT, reverse: CACGACACGAGCTGACG). The thermal cycling conditions included initial denaturation for 10 min at 95 °C followed by 30 cycles of 30 s at 95 °C, 30 s at 57 °C, and 30 s at 72 °C. PCR amplicons (≈300 bp) were purified and libraries were prepared using the NEBNext Fast DNA Library Prep Set for Ion Torrent (New England BioLabs) and Ion Xpress Barcode Adapters 1–96 Kit (ThermoFisher Scientific). Libraries were consequently pooled, with their equimolar concentration determined with a KAPA Library Quantification Kit (KAPA Biosystems). The sequencing template was prepared in a One Touch 2 instrument using an Ion PGM OT2 HiQ View Kit (ThermoFisher Scientific). HTS was performed in an Ion Torrent PGM platform with an Ion 316 Chip Kit v2 BC (ThermoFisher Scientific) using an Ion PGM Hi-Q View Sequencing Kit (ThermoFisher Scientific), according to the manufacturer’s protocols.

### 2.4. Bioinformatic Analysis

Raw sequences retrieved from the Ion Torrent Software Suite in the fastq format were processed using the software Qiime2, which was specifically designed for microbial ecology [25]. Sequences were quality filtered, trimmed, dereplicated, and denoised using DADA2 software, and chimeras were removed [26]. Taxonomy was assigned with a VSEARCH-based consensus classifier against Greengenes database version 13_8 [27]. Sequences were rarefined at a minimum sequencing depth of 5221 reads (Appendix A). The analysis of bacterial diversity was assessed through alpha diversity (Chao1, evenness, Faith’s phylogenetic diversity, and Shannon index) and beta diversity (Jaccard’s distance metric) using the Qiime2 pipeline. EMPeror was used for the visualization of principal coordinates analysis [28]. Boxplots were created using the libraries numpy 1.18.3, pandas 1.0.3, matplotlib 3.2.1, and seaborn 0.10.1 in Python version 3.8.2. Sequence information was deposited in the Sequence Read Archive under the accession number PRJNA645883.

### 2.5. Statistical Analysis

Alpha diversities among patient groups were compared by non-parametric tests using either the Mann–Whitney *U* test for two groups or Kruskal–Wallis *H* test for multiple groups. Statistical *p*-values and *q*-values with Benjamini–Hochberg false discovery rate correction are reported. In the same manner, *p*-value and *q*-value correction are shown for PERMANOVA with 999 permutations on beta diversities among the studied groups. Additionally, the PERMDISP test was done to support the PERMANOVA results. Differential abundance analysis was performed on the Huttenhower Galaxy Server using linear discriminant analysis effect size with the standard parameters [29].

## 3. Results

### 3.1. Study Group Characterization and Clinical Response

This study included 16 patients, 8 men (M) and 8 women (F), suffering from left-sided UC who received either 5-ASA (*n* = 8) or FMT (*n* = 8) enema treatment. Basic patient characteristics are summarized in Table 1. Samples were collected before the start of treatment (baseline) and at multiple time points during the treatment; in total we received 21 samples from 8 patients of 5-ASA group at three sampling points and 39 samples from 8 patients of FMT group at five sampling points (Appendix A). On the basis of their Mayo score (disease activity index) at weeks 6 and 12, we divided the subjects into responders and non-responders. Treatment responders were considered subjects with an achieved clinical remission, defined as a Mayo score ≤ 2, with no subscore > 1, which was the primary endpoint. Secondary endpoints were (a) clinical response, defined as a reduction in Mayo score of at least 2, and (b) endoscopic remission defined as an endoscopic Mayo score of 0. Treatment non-responders thus did not achieve the primary endpoint, but they may or may not have achieved secondary endpoints. In the FMT group, 37.5% (three out of eight subjects) reached the primary endpoint, 62.5% (five out of eight) had a clinical response, and 12.5% (one out of eight subjects) reached endoscopic remission. In the 5-ASA group 50% (four out of eight subjects) reached the primary endpoint, 62.5% (five out of eight subjects) had a clinical response, and 37.5% (three out of eight subjects) reached endoscopic remission. In the 5-ASA group, men and women were distributed equally between responders and non-responders, while the effect of FMT therapy was less evident in women, as only one female reached clinical remission. The fecal sample used for FMT treatment was obtained from one donor who underwent rigorous selection criteria and screening investigations. The inclusion and exclusion criteria of the patients and donor are described in Appendix A, respectively. No adverse events were reported during the treatment and 6 weeks after treatment.

### 3.2. Alpha and Beta Diversity

Alpha diversity, which evaluates the species richness and evenness; Faith’s phylogenetic distance; and Shannon diversity showed no significant differences between the FMT and 5-ASA treatment groups, nor between the responder and non-responder subgroups inside each cohort. Additionally, at the baseline, responders could not be distinguished from non-responders. A non-significant increase was detected in Shannon diversity index, for both responders and non-responders, 2 weeks after both therapy types compared to the baseline (Appendix A). A higher Shannon index was still observed when more samples from different sampling points after therapy initiation were included in the calculation, indicating that FMT and 5-ASA can, to a certain extent, influence the microbial alpha diversity of UC patients.

Beta diversity, which evaluates the similarity of bacterial communities among samples, was assessed using Jaccard’s non-phylogenetic distance matrix. As early as 2 weeks after the therapy initiation, we could differentiate responders from non-responders in both the FMT group (PERMANOVA *p* = 0.001, PERMDISP *p* = 0.100) and 5-ASA group (PERMANOVA *p* = 0.003, PERMDISP *p* = 0.099). Figure 1 shows the separation of responders and non-responders within the 5-ASA treatment group (21 samples) and FMT treatment group (39 samples) resulting from the analysis of samples at the baseline and at different sampling points after therapy initiation. The separation of FMT cluster centroids was documented by PERMANOVA *p* = 0.001, however, the results can be partially influenced by high intergroup variability (PERMDISP *p* = 0.022). The separation of 5-ASA cluster centroids is supported by PERMANOVA *p* = 0.001 (PERMDISP *p* = 0.053). At the baseline, responders could not be differentiated from non-responders in the FMT group (PERMANAOVA *p* = 0.066, PERMDISP *p* = 0.336) nor in the 5-ASA group (PERMANOVA *p* = 0.223, PERMDISP *p* = 0.414). As for the similarity of subjects to the donor after FMT, Appendix A shows that FMT responders were closer (more similar) to the donor than FMT non-responders (Mann–Whitney *U* test, *p* = 0.00003).

### 3.3. Taxonomical Composition

In total, 9 phyla, 142 genera, and 184 species were detected in the samples of UC patients. Firmicutes (41–94%) were detected as the dominant phylum in all samples, regardless of treatment, except for one sample (19%) from the FMT group at the baseline, in which Proteobacteria (52%) were flourishing. The second most abundant were Actinobacteria (1–38%), and/or Bacteroidetes (1–37%), as shown in Figure 2. Firmicutes were mainly represented by the order Clostridiales; Bacteroidetes were mainly represented by the order Bacteroidales; and in Actinobacteria, the order Bifidobacteriales predominated. Other phyla including Fusobacteria, Tennericutes, Acidobacteria, Planctomyceles, and TM7 were detected with low frequencies (≤0.4%).

The donor stool was dominated by Firmicutes, with a prevalence of the families Lachnospiraceae (67%) and Ruminococcaceae (17%). The relative abundance of Actinobacteria (1%) and Bacteroidetes (2%) was quite low, represented by the family Coriobacteriaceae and the families Prevotellaceae and Bacteroidaceae, respectively. *F. prausnitzii* was present with a frequency of 3% in the stool of the donor.

Venn diagram analysis shows the number of genera shared between the donor and the subjects of the FMT group before treatment (Figure 3A), and at all sampling points after the start of the treatment (Figure 3B). Collectively, stool samples of non-responders after FMT contained the highest number of unique genera (62), while for FMT responders, the diagram shows only four unique genera (Figure 3B). Such a big difference indicates that non-responders still harbored a high amount of their original unique genera. After FMT, responders retained 50 original genera from their previously determined 67 genera (75%) and accepted 17 new genera. In contrast, non-responders retained 61 of their original 65 genera, indicating that 94% of genera remain unchanged, which could be one of the reasons for their therapy non-responsiveness. Interestingly, this shift was not evident at week 2 after the beginning of therapy, when both subgroups kept all their original bacterial settlement, which was only enriched by several new genera (data not shown). This fact indicates that at this time point only new genera occurred in the community with no bacterial replacement, meaning the new community structure had not yet been established.

Linear discriminant analysis effect size (LEfSe) was applied to the set of samples to determine bacterial taxa with significantly different levels of abundance in responders and non-responders in both treatment groups. No differentially abundant taxa were determined between responders and non-responders inside the 5-ASA treatment group. In the FMT group, collectively, 26 significantly different taxa were identified between responders and non-responders (linear discriminant analysis score > 2), however, at baseline, the therapy responsiveness was not found to be significantly associated with any bacterial taxa. On the family level, Lachnospiraceae, Ruminococcaceae, and Clostridiaceae of phylum Firmicutes, and Bifidobacteriaceae and Coriobacteriaceae of phylum Actinobacteria were significantly increased in the subgroup of FMT responders. At the genus level, *Faecalibacterium* and *Blautia* (Ruminococcaceae), *Coriobacteria*, *Collinsela*, *Slackia* (Coriobacteriaceae), and *Bifidobacterium* (Bifidobacteriaceae) were significantly more abundant in FMT responders. In the subgroup of FMT non-responders, the families Lactobacillaceae, with an increased *Lactobacillus* genus, and Christensenellaceaea, both of the phylum Firmicutes, and Paraprevotellaceae of the phylum Bacteroidetes, with an increased *Paraprevotella* genus, were present with a significantly higher frequency. At the genus level, *Oscillospira* (Ruminococcaceae) and *Odoribacter* (Porphyromonoadaceae) also had a higher abundance in FMT non-responders (Figure 4). For more detailed analysis at the species level, refer to Appendix A.

To elucidate the microbiome alteration during the first 4 weeks, we performed the LEfSe analysis of samples collected at week 2 and 4 after FMT. At week 2, FMT responders showed significantly increased abundance of unclassified Clostridiaceae, and unclassified *Prevotella*, *Slackia*, and *Turicibacter* compared to non-responders (Appendix A). At week 4, significantly increased *Fecalibacterium prausnitzii* was detected in responders and unclassified genus of Christensenellaceaea and *Paraprevotella* were detected in non-responders (Appendix A). These preliminary results indicate the bacterial abundance changes over time and the community composition instability between week 2 and 4. Except *Turicibacter*, all significantly increased taxa were present in the overall LEfSe analysis (Figure 4).

## 4. Discussion

In this monocentric work, 16 patients with active left-sided UC lasting more than 3 months were enrolled to study the effect of FMT in comparison with 5-ASA treatment administrated by enema. Major differences exist between these therapies, as FMT has a direct influence on the microbiota composition, while 5-ASA should act as an anti-inflammatory agent. 5-ASA compounds, usually administrated orally, are a highly effective treatment for UC [30,31]. The delivery systems designed for conveying 5-ASA to the colon include various pH-dependent polymers, microgranules encapsulated into ethyl cellulose, or azo-bound derivatives. However, none of these compounds are as effective as the topical formulations [32]. However, patients do not easily accept local therapy, and long-term treatment may not be acceptable to many patients [33]. In patients with irritable bowel syndrome, it was reported that 5-ASA reduces the amount of fecal bacteria quite drastically, by over 40% [34]. Microbiota changes after 5-ASA treatment have been found in mucosa, and to a lesser degree also in feces [5]. In mucosa, an inverse correlation with disease severity was shown for *F. prausnitzii*, other short-chain fatty acids producers, and many more bacteria after 5-ASA treatment [4,5]. A high mucosal 5-ASA concentration was related to a reduced abundance of pathogenic bacteria in mucosa such as Proteobacteria, and increased abundance of several favorable bacteria such as *Faecalibacterium*. In feces, *Prevotella* and *Sutterella* were decreased upon an increase in 5-ASA in mucosal tissue [5]. *Sutterella* is thought to contribute to UC pathogenesis by its ability to degrade mucosal protective immunoglobulin A (IgA) antibody [35,36]. We did not find any significant differences in alpha bacterial diversity between 5-ASA treatment responders and non-responders. However, significant Jaccard distances between these subgroups showed some effect of mesalazine application in responders, which is also supported by a Mayo score index ≤ 2. Perhaps analysis of mucosal microbiome, which is at the site of mesalazine’s action, would reveal more profound changes; however, more research is needed to elucidate this effect.

FMT as an alternative treatment of UC patients is attracting increased attention, and the number of studies has been growing steadily in recent times. Although several trials have given promising results [11,12,37,38] (see systematic reviews [10,39,40]), many unanswered questions remain that require further research.

The increased microbiota diversity reported by several authors in UC patients after FMT from multi-donor blended stool (2–7 donors) [11,12,13,14,15] was not found in our study using one donor for all patients. The finding reported in this work of no significant differences in Shannon diversity in FMT responders and non-responders between the baseline and any other sampling point after therapy initiation is in agreement with the studies of Damman et al. [17] and Kump et. al. [16], who, however, described temporal changes in mucosal, but not fecal samples 7 days after FMT. Both studies [16,17] used individual stool donation, which means that each donor provided a stool for one, at most two recipients. Tian et al. [41] even described non-significantly decreased Shannon and Chao indices after FMTs, but still observed positive clinical outcomes and improved symptoms in the patient with UC. The same authors [41] did not find differences in beta diversity; however, in our study, statistically significant results were obtained from pairwise PERMANOVA analyses. Jaccard distances between samples revealed the separation of responders and non-responders after FMT treatment, however, the high variability of samples within the FMT group analyzed in this work has to be taken into consideration. Furthermore, a higher similarity of responders with the donor was found in the FMT group. This finding is in agreement with several studies [10,15,16,18,42], however, not all of them reported the correlation between the bacterial shift towards the healthy donor and clinical response [16].

The efficacy of FMT is here further supported by the lower proportion of the original genera maintained by responders, and the relatively high proportion of original genera maintained by non-responders. In FMT non-responders, the number of unique genera was high, which indicates unsuccessful restoration of the disrupted microbiome, inability to replace a certain proportion of original genera, and possibly resistance of some genera to this type of intervention. From a statistical point of view, these results could however be influenced by the higher number of FMT non-responders (five subjects) compared to responders (three subjects), which could increase the diversity within the subgroup of non-responders. Shi et al. [10] emphasized, on the basis of 25 trials using FMT treatment for UC, that patients sharing increased bacterial similarity with the donor can exhibit different clinical outcomes, and thus the mere presence of healthy microbiota is not sufficient to achieve a positive effect of FMT. In contrast, according to Kump et al. [38], the taxonomic composition of the donor’s intestinal microbiota is a major factor influencing the efficacy of FMT in UC patients. Our preliminary results could also raise a question of a gender factor role in FMT efficacy. We noted that there was only one female responding to FMT treatment by stool donated by a man. This finding, however, must be assessed with great caution due to the small number of subjects analyzed in this study. Nonetheless, it is increasingly apparent that sex is one of the important variables affecting the gut microbiota [43,44,45]. The FMT animal model study even showed that female recipients lost significantly more weight after receiving the male microbiota compared with the weight after receiving the female microbiota [46]. Sex or gender factors should not be ignored by researchers; however, this association has not yet been sufficiently investigated.

The “proper” taxonomical composition of a healthy microbiome is, however, still unknown, largely because of the huge inter-individual variability across the entire population. Hence, the selection of a good donor is quite challenging, although there is some evidence that certain donors can be better than others with respect to FMT efficiency. Nevertheless, we still do not have specific criteria for donor selection. Literature data suggest that certain bacterial taxa in the donor microbiota seem to be associated with treatment response to FMT, especially *Akkermansia muciniphila* [38], butyrate-producing *F. prausnitzii* [20,38], *Roseburia intestinalis* [6] and *Roseburia faecis* [20], *Butyrivibrio crossotus*, *Anaerobutyricum hallii* (reclassified *Eubacterium hallii*) [6], and some members of the families Ruminococcaceae [37,38] and Lachnospiraceae [37], which have been identified as favorable bacteria for UC treatment. On the other hand, some pro-inflammatory bacteria of the phylum Proteobacteria and *Ruminococcus gnavus* are thought to be harmful [6]. Recently, the term “super-donor” has been proposed to describe donors whose stools result in significantly more successful FMT outcomes than the stools of other donors [37,47], and the bacteria mentioned above should be part of their intestinal microbiota.

We have identified several of the beneficial bacterial taxa in the microbiome of the donor who provided the sample for the FMT performed here, including *F. prausnitzii* (3%), members of the Ruminococcaceae family (20%), and a high percentage of bacteria of the Lachnospiraceae family (48%), which were also present in the stool of the most successful donor of the Moayyedi study [37].

Two Lachnospiraceae clusters were identified here as significantly increased in responders after FMT therapy, similar to the findings of Kump et al. [16] and Angelberger et al. [20]. Many Lachnospiraceae members have been detected in the human intestine [48] and some exhibit important hydrolytic activities (e.g., xylanase, β-xylosidase β-galactosidase, α- and β-glucosidase, α-amylase, pectin methyl-esterase, pectate lyase, N-acetyl-β-glucosaminidase) [49]. Lower amounts of Lachnospiraceae were previously reported in a subject suffering from UC [50], however, Vacca et al. [51] pointed out the increased abundance of Lachnospiraceae in the intestinal lumen of subjects with different diseases, thus indicating the possible controversial role of this taxon. Nevertheless, several genera are known for their positive effect on health, especially butyrate-producing strains of *Butyrivibrio*, *Roseburia*, *Anaerostipes*, or *Coprococcus* [52].

FMT also significantly induced a higher abundance of the family Ruminococcaceae in responders and two important members of this taxon, *F. prausnitzii* and *Blautia*. Both these genera are abundant in the human intestinal microbiota of healthy adults [53], while reduced levels of *F. prausnitzii* and/or *Blautia* have been reported in UC individuals [54]. The positive effect of these bacteria is attributed to the production of butyrate. Butyrate plays a major role in gut physiology, with numerous beneficial effects on health through anti-inflammatory activities in the colonic mucosa, protection against pathogen invasion, modulation of the immune system, and reduction of cancer progression [55,56]. *F. prausnitzii* has even been suggested to constitute a marker of a healthy gut [57,58]. The increased levels of *F. prausnitzii* after FMT found in this study have been also reported by Chen [14] and Fuentes [6], indicating a positive influence of FMT. However, we did not find any data about the effect of FMT on *Blautia* in UC patients in the corresponding literature.

Ten taxa elevated in FMT responders belonged to the phylum Actinobacteria, with significantly increased Bifidobacteriaceae and Coriobacteriaceae at the family level. Bifidobacteria are believed to exert positive health benefits on the host via their metabolic activities [59] and have been successfully used in UC patients as a probiotic treatment, resulting in remission throughout the trial [60]. As the family Bifodobacteriaceae was found to be reduced in most IBD patients [61], we can consider the increase found in this study to be favorable. *Collinsella* and *Slackia* are both members of the family Coriobacteriaceae. In the gut, Coriobacteriaceae perform important functions, such as the conversion of bile salts and steroids and the activation of dietary polyphenols [62]. *Slackia* was found in low numbers in human feces of healthy subjects, and it is thought to play an important role in gut health [63]. Some species are involved in equol production (daidzein-to-equol conversion), which is exclusively a product of the intestinal bacterial metabolism of dietary isoflavones. Equol possesses estrogenic activity and is superior to all other isoflavones in terms of antioxidant activity [64]. We did not find any literature data correlating the abundance of this genus with ulcerative colitis. However, *Collinsella* was found at lower frequencies in children and adults with UC and Crohn’s disease (CD) [65,66], and in patients with irritable bowel syndrome [67]. A significant decrease in the family Coriobacteriaceae was observed in stool samples from patients with CD [68], however, some authors consider *Collinsella* to be a pathobiont because its occurrence has been associated with type 2 diabetes [69], the progression of insulin resistance during pregnancy [70], rheumatoid arthritis [71], and cholesterol metabolism [72]. In the CACO-2 epithelial cell line, *Collinsella* increased gut permeability by reducing the expression of tight-junction proteins and inducing the expression of interleukin 17 (IL-17) network cytokines, which are frequently involved in inflammatory diseases [71]. The assessment of the increased levels of this bacterium in UC patients after FMT is thus quite complicated, and the clinical relevance of the members of family Coriobacteriaceae for gut health will certainly receive increased attention.

Several taxa were found at higher frequencies in the patients with UC who were classified as FMT non-responders. Surprisingly, the increased taxa of the phylum Firmicutes are mostly associated with a healthy gut. Christensenellaceae (order Clostridiales), a recently described family, seem to be a highly heritable and important player in human health [73]. Members of this family have been associated with healthy dietary habits [73] and with human longevity [74], were negatively correlated with serum lipids [75], were enriched in individuals with low body mass index [76], and increased after diet-induced weight loss [77]. Christensenellaceae were consistently depleted in individuals with Crohn’s disease [78,79,80] and UC [68,81,82], which is contrary to our results. As the relative abundance of Christensenellaceae was found to be increased with age [73,83], the possible association with the higher median age of FMT non-responders (42 years) compared to responders (28 years) in this study should be noted. Mancabelli et al. [84] reported Christensenellaceae to be one of the taxa considered a signature of a healthy gut, and cultured members of Christensenellaceae have potential as therapeutic probiotics for the improvement of human health [85]. However, the functional role of Christensenellaceae in the gut is still not understood.

*Oscillospira* of the family Ruminococcaceae is a common genus found in about 60% of all individuals in several large metagenomic datasets of the fecal human microbiota [86] and is also thought to have positive contributions for human health due to its putative butyrate production [87]. A meta-analysis of five microbiota studies in patients with IBD indicated a significantly reduced incidence of *Oscillospira* in patients with CD [88], but we did not find any information about the abundance of this genus in UC patients.

The role of Lactobacilli, which are generally recognized as beneficial for human health for their probiotic effects, is not so unequivocally clear in gut inflammatory diseases. The proportions of these bacteria are frequently either positively or negatively correlated with human disease and chronic conditions [89]. Several studies show that *Lactobacillus* is depleted in IBS patients [90,91], decreased in UC patients [65], and increased in CD patients [92]. It is not known whether *Lactobacillus* participates in the disease or has simply adapted to survive the pro-inflammatory gut environment. Additionally, the effect of probiotic *Lactobacillus* consumption differs, only resulting in improved clinical symptoms in IBS and UC patients [93,94,95]. Knowledge about metabolic differences among strains and/or species of Lactobacilli could be useful to evaluate variations in the involvement and contributing factor of this genus in different diseases [89].

Two genera elevated in FMT non-responders belonged to the phylum Bacteroidetes. *Odoribacter* (family Porphyromonoadaceae, order Bacteroidales) is a butyrate-producing member of the human intestinal microbiome [96] and its proper abundance is crucial for a healthy gut [97]. A reduced frequency of this genus was found in patients with the most severe form of UC (pancolitis) [97] and CD patients [98]. On the other hand, higher levels of *Odoribacter* were correlated with an improved state of health with CD [99]. As this genus is thought to play a positive role against gut inflammation, the increased abundance found in our work can be evaluated as beneficial. This, however, cannot be deduced for *Paraprevotella* (family Prevotellaceae, order Bacteroidales). This genus is characterized by the production of succinic acid [100], which can be associated with microbiome dysbiosis and intestinal inflammation [101]. Normally, succinate is detected at low concentrations in the gut lumen because of its rapid conversion into propionate; however, several studies found a higher concentration of succinate in IBD patients [102], and an association between succinate accumulation in the gut lumen and microbiota disturbances has recently emerged [101,103]. Here, we have to emphasize that *Odoribacter* is a succinate-consuming bacterium, and its increased abundance may be theoretically related to *Paraprevotella* succinate production. The role of succinate in inflammatory processes within the gut is however unclear [101], and more research is required to elucidate the implications of succinate on intestinal inflammation.

On the basis of our results, we can conclude that 5-ASA topical treatment resulting here in 50% remission is an adequate type of therapy, however, we were unable to identify the fecal bacterial changes associated with the cure response. FMT is in our opinion a promising treatment method for patients suffering from UC. Despite the only 37.5% clinical remission achieved by FMT in this study, the increased abundance of butyrate-producing bacteria indicates a positive bacterial shift. Our data indicate that the presence or increased abundance of these beneficial bacteria is not a sufficient factor to achieve improved clinical outcomes. The replacement of certain intestinal bacteria with health-promoting genera seems to be an indispensable condition for successful FMT therapy. This study has some limitations, however. The small number of patients enrolled in this study, partially caused by the focus on subjects with left-sided UC and the incomplete set of patient stool samples, mean that this work should be treated as a preliminary study, and thus further evaluation is needed. A larger cohort of patients including control groups could be used to further elucidate the changes in microbiota after FMT and to evaluate the clinical response. The results of such studies can help to understand which bacterial groups are beneficial, but also transferable from a donor to recipient. FMT has the potential to be established as an effective and safe treatment for UC patients, especially when standard treatment has failed. The research in this field is still limited, with many problems that need to be solved and many questions that need to be answered in order to confirm the efficacy of this alternative approach supported by clinical outcomes.

## Figures and Tables

**Figure 1 cells-09-02283-f001:**
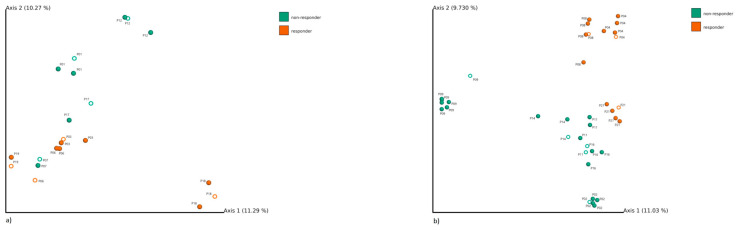
Principal coordinate analysis showing Jaccard’s distance matrix between responders (orange) and non-responders (green) for (**a**) aminosalicylates (5-ASA) group (8 patients) and (**b**) fecal microbiome transplantation (FMT) group (8 patients). Samples collected before therapy are shown as hollow spheres (8 samples per each 5-ASA and FMT group), samples collected after therapy initiation are shown as full spheres (13 samples for 5-ASA and 31 samples for FMT group); samples belonging to one patient are described with the same number.

**Figure 2 cells-09-02283-f002:**
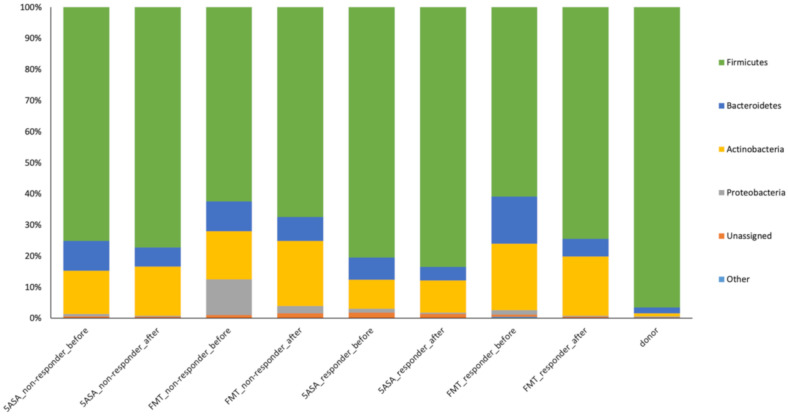
Relative abundance of fecal bacteria at the phylum level in all 60 samples from 16 ulcerative colitis (UC) patients with active left-sided colitis and the donor grouped by therapy type (FMT, 5-ASA), responsiveness (responder, non-responder), and time point of a sample collection (before therapy or at multiple time points after therapy initiation). Fusobacteria, Tennericutes, Acidobacteria, Planctomyceles, and TM7 with low relative abundance are summarized as “Other”.

**Figure 3 cells-09-02283-f003:**
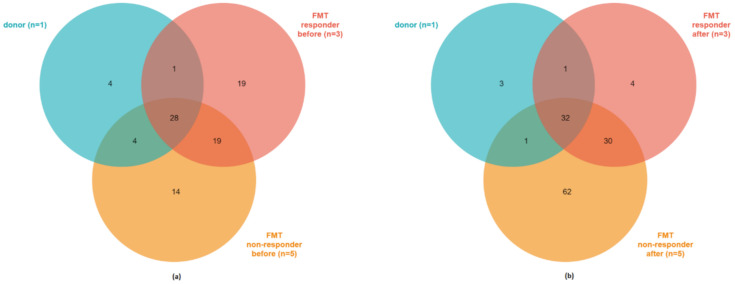
Venn diagram analysis of bacterial genera in healthy donor and patients with active left-sided UC treated with FMT (**a**) before the therapy (baseline) and (**b**) at all sampling points after the start of therapy. The number in each region represents genera shared between the sample groups (overlapping regions) or genera unique for the sample group. Number of subjects inside the group is indicated in parentheses.

**Figure 4 cells-09-02283-f004:**
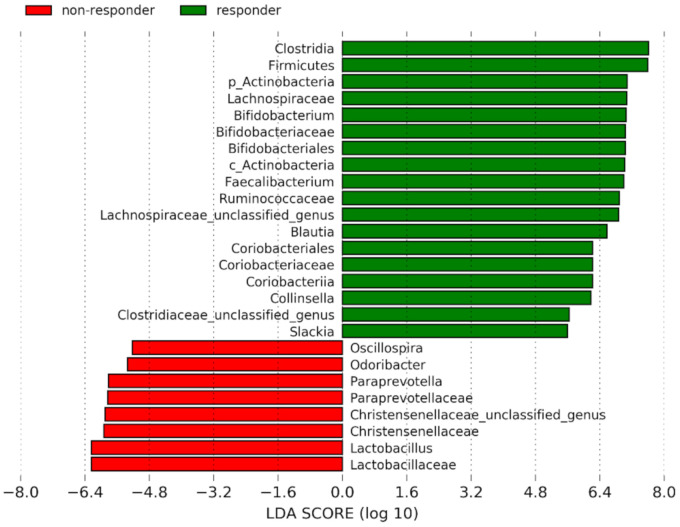
Linear discriminant analysis (LDA) scores of responders and non-responders in the FMT group of patients with active left-sided UC on different taxonomical levels (phylum, class, order, family, genus) for all sampling points including baseline. Actinobacteria phylum and class are distinguished using shortcuts p and c, respectively.

**Table 1 cells-09-02283-t001:** Patient study group characteristics before therapy.

Characteristics	5-ASA Group (*n* = 8)	FMT Group (*n* = 8)
Male/female	4:4	4:4
Age median (range)	40 (31–66)	37.5 (28–62)
Number of samples provided per patient median (range)	3 (2,3)	5 (3–7)
Mayo score median (range)	5.5 (4–9)	5.5 (3–9)
Endoscopic Mayo score median (range)	2 (2–2)	2 (2–2)
CRP mg/L median (range)	0.85 (0.2–10.4)	1.35 (0.2–13.2)
WBCC ×10^9^ median (range)	5.5 (4.0–8.4)	7.9 (6.2–9.0)
Patients on thiopurines	1	1
Patients on corticosteroids	2	0
Patients on mesalazine	8	6

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
