# Peer review of "Gut Microbiome Changes in Patients with Active Left-Sided Ulcerative Colitis after Fecal Microbiome Transplantation and Topical 5-aminosalicylic Acid Therapy"

_cells, 2020, doi:10.3390/cells9102283_

Round 1

Reviewer 1 Report

In this study, the authors aimed to determine the influence of aminosalicylates (5-ASA) topical treatment and fecal microbial transplantation (FMT) treatment on the fecal bacterial community in patients with left-sided ulcerative colitis (UC) and to evaluate the consequent clinical response. The authors concluded that 5-ASA topical treatment resulting here in 50% remission is an adequate type of therapy; FMT is a promising treatment method for patients suffering from UC, despite the only 37.5% clinical remission achieved by FMT. Their data also indicated that the presence or increased abundance of these beneficial bacteria is not a sufficient factor to achieve improved clinical outcomes.

Comments

This is an interesting study. The reviewer has some concerns as follows:

  1. There are only 8 patients for FMT and 8 patients for 5-ASA. The sample size may be a limitation for this study. The authors may need to mention and discuss this issue.
  2. The number ratio of male : female patients are 4:4 in each group. However, the authors did not show and discuss the results of male : female. It is interesting for the involvement of gender factor.
  3. The authors should provide the detailed information/characteristic for FMT donor.
  4. In the Title of this manuscript, the full name for 5-ASA is recommended.

Reviewer 2 Report

The researchers have evaluated the efficacy of faecal microbial transplant (FMT) by enema and 5-ASA by enema on the faecal microbiome of sixteen patients [FMT (n=8), 5-ASA (n=8)] with active left-sided ulcerative colitis. The faecal microbiota was monitored by 16S rRNA high-throughput sequencing, and standard clinical indices were used to assess treatment efficacy

 FMT promoted remission in 37.5 % (3/8) of patients that were linked with significant increases in the relative abundance of Firmicutes, and Actinobacteria, in particular Faecalibacterium, Blautia, Coriobacteria, Collinsela, Bifidobacterium and Slackia. These changes were not evident in FMT non-responders. 5-ASA was more effective and led to clinical remission in 50 % (4/8) of patients, but there was no correlation to changes in the faecal microbiome. The authors conclude that FMT may a promising non-drug treatment for UC.

Although the numbers of patients were low the data does, in fact, indicate that FMT was inferior to 5-ASA both in the ability to promote remission or to elicit any amelioration to those defined as non-responders. If FMT has a role in therapy, it would be as an adjunct to other therapies. Is there any indication that FMT might have been helpful in the treatment of 5-ASA non-responders?

A major complication to this study is how the microbiome was analysed after initiation of treatment. Faecal samples were collected at 2, 4, 6 and 12 weeks but the data has not been analysed as a time course. Rather, the data from each individual sample has been pooled to give an overall comparison of the before and after microbiome [Figure 2-4]. One cannot gauge whether the reported differences with FMT were transient (during the period in which transplant was given) or persistent (at 12 weeks) which was 6 weeks after the last dose. A time course or data for 12 weeks would give a much better indication of any long-lasting effects of FMT on the faecal microbiome.

Ln 90               ‘reached by our methods’. Presumably, colonoscopy or enema.

Ln 96               Were the patients hospitalised during the first 6 weeks of study or did they attend clinics?

Ln 105             During week 1-6 were the samples collected prior to dosing with FMT.

Ln 105             ‘Sample collection is not fully complete’. Need to report (supplemental) how many samples were collected from each patient and when.

Ln 107             ‘The donated stool for FMT’. Was this a single or multiple samples? How was sampled stored and processed prior to use?

Ln 162             Are patients that are non-responders classed as those who only show secondary clinical endpoints.

Ln 165-169     Check the numbers. At present, the 5-ASA group has more than 8 participants.

Ln 227-240     Does Venn (b)represent all the samples collected after initiation of treatment.

Ln 248-263     It looks that this data is from samples collected after initiation of treatment. If so, was there any difference between responders and non-responders prior to the start of treatment?

Ln 270-302     This is repeating much of the introduction and should be added to the introduction if required.

Ln 328-330     I am unsure how the number of non-responders affects the data unless they were included the overall analysis?

Figure 2           Is data for responders and non-responders after the start of study taken at the same study timepoint?

Figure 3-4       Is data after initiation of treatment a summation of all time points.

Reviewer 3 Report

In the manuscript, the authors explored the potential changes of gut microbiota in the patient with ulcerative colitis after the treatment of 5-ASA and/or fecal microbial transplantation (FMT), and showed that FMT treatment increased relative abundance of the family lachnospiraceae, etc. Major concerns: 1. The number of patients included in the manuscript is too small, which makes the results/conclusion are not convincing. 2. Whether was the gut microbiota of the donor evaluated? How did the author determine whether the donor's gut microbiota was normal or not? 3. How does the gut microbiota impact the clinical efficacy of 5-ASA and FMT? 4. The authors did not show the changes of gut microbiota species between 5-ASA responders and non-responders.

Round 2

Reviewer 2 Report

The authors have dealt with all raised queries in a satisfactory manner.

Reviewer 3 Report

The authors have addressed the majority of my concerns.